# VT-PLUG: Integrating Visual Task Plugins with Unified Instruction Tuning

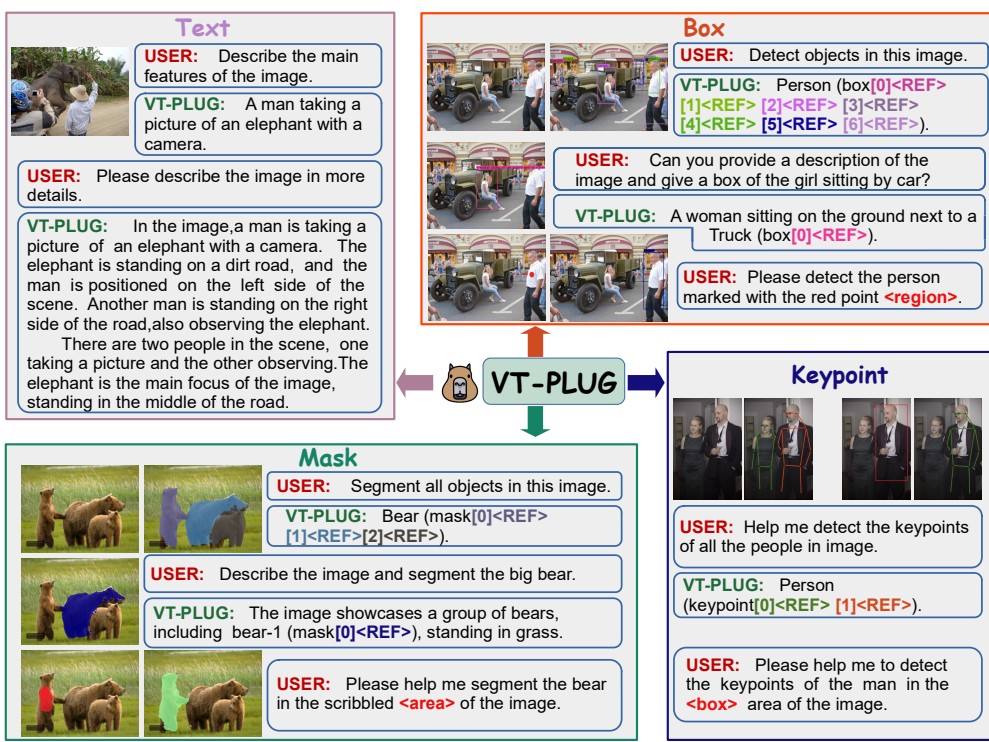

Figure 1: **Overview of Visual Tasks Supported by VT-PLUG.** VT-PLUG supports user-provided visual inputs such as points, boxes, scribbles, and masks, while enabling the decoding of visual contents into formats like boxes, keypoints, and masks. The combination of these input and output formats facilitates a wide range of visual tasks.

## Abstract

Multimodal Large Language Models (MLLMs) demonstrate robust zero-shot capabilities across diverse vision-language tasks after training on mega-scale datasets. However, dense prediction tasks, such as semantic segmentation and keypoint detection, pose significant challenges for MLLMs when represented solely as text outputs. These challenges often necessitate task-specific visual decoders, leading to the underutilization of MLLMs' multi-task potential. In this work, we propose **VT-PLUG**, a novel framework that leverages modular visual components as scalable plugins for a variety of visual applications. During the joint training of vision-language tasks with varying prediction densities, we propose a **Visual Decoding Chain-of-Thought (VD-CoT)** mechanism to prevent task conflicts. VD-CoT requires the model to predict the current task's recognition entities, decoding unit type, and other specific details, while also providing learnable queries for precise decoding. Additionally, we construct **Visual-Task Instruction Following Dataset (VT-Instruct)**, a large-scale multi-task dataset

containing over 100 million multimodal dialogue samples across 25 task types. Beyond text inputs and outputs, VT-Instruct incorporates various visual prompts such as point, box, scribble, and mask, and generates outputs composed of text and visual units like box, keypoint, depth and mask. The combination of different visual prompts and visual units generates a wide variety of task types, expanding the applicability of VT-PLUG significantly. The source code, dataset and demo will be released at `https://anonymous.4open.science/r/VT-PLUG`.

# 1 INTRODUCTION

Multimodal Large Language Models (MLLMs) demonstrate excellent performance in tasks such as visual question answering and scene understanding (Liu et al., 2024; Alayrac et al., 2022; Tai et al., 2024). Despite these achievements, typical MLLMs primarily understand input and generate responses with text, which limits their ability to perform fine-grained visual localization. As a result, they struggle to make significant contributions in real-world applications such as autonomous driving, robotics, and medical diagnosis.

In this work, we introduce **VT-PLUG**, a novel framework that utilizes modular visual components as scalable plugins for visual tasks with varying degrees of prediction density. To support common visual tasks, we designed Visual Prompt (VPT) encoding plugins alongside point, box, mask, and keypoint decoding plugins. These various visual plugins can be combined to tackle complex multimodal tasks or easily extend to new visual applications. For instance, the VPT encoding plugin can pair with different decoding plugins to generate visual decoding units for diverse user interaction modes. A combination of box and keypoint plugins also enables efficient multi-person pose estimation. As shown in Table 1, compared to existing MLLMs which focus on fine-grained visual localization and understanding, VT-PLUG offers greater flexibility and can accomplish a wider variety of visual tasks.

VT-PLUG utilizes tokens generated by MLLMs as learnable queries for decoding. Therefore, we propose the **Visual Decoding Chain-of-Thought (VD-CoT)**, which requires the model to output special tokens for VT-PLUG decoding, as well as other task-related information during the generation process. VD-CoT decomposes the response generation process into two stages: Visual CoT and Decoding Triplets. During the Visual CoT stage, the model extracts the categories and quantities of visual entities to be decoded based on the image and prompt information, and determines the corresponding decoding types. In the Decoding Triplets stage, visual-related decoding information is divided into three types: `Phrase`, `Unit`, and `<REF>`, forming multiple triplets. `Phrase` represents the category of visual entities, `Unit` indicates the decoding type, and `<REF>` corresponds to the learnable queries for VT-PLUG. Since the `<REF>` token appears at the end, it effectively leverages relevant information from the previously generated content.

Table 1: Comparisons of recent MLLMs and their capabilities in performing downstream tasks.

| Model | End-End Model | Extend -ability | Visual Understanding | | Referring Expression | | | Interactive Grounding (IG) | | Grounded Conversation Generation (GCG) | | Open Vocabulary Identification | | Keypoint Detection |
|---|---|---|---|---|---|---|---|---|---|---|---|---|---|---|
| | | | VQA | Caption | RES | REC | REG | Mask | Box | Mask | Box | OVS | OVD | |
| LLaVA (Liu et al., 2024) | ✔ | - | ✔ | ✔ | - | - | - | - | - | - | - | - | - | - |
| BuboGPT (Zhao et al., 2023) | - | - | ✔ | ✔ | - | ✔ | - | - | - | - | ✔ | - | - | - |
| Kosmos-2 (Peng et al., 2023) | ✔ | - | ✔ | ✔ | - | ✔ | ✔ | - | - | - | ✔ | - | - | - |
| Shikra (Chen et al., 2023b) | ✔ | - | ✔ | ✔ | - | ✔ | ✔ | - | - | - | ✔ | - | - | - |
| MiniGPT-v2 (Chen et al., 2023a) | ✔ | - | ✔ | ✔ | - | ✔ | ✔ | - | - | - | ✔ | - | - | - |
| NExT-Chat (Zhang et al., 2023a) | ✔ | - | ✔ | ✔ | ✔ | ✔ | ✔ | - | - | ✔ | ✔ | - | - | - |
| Ferret (You et al., 2023) | ✔ | - | ✔ | ✔ | - | ✔ | ✔ | - | ✔ | - | ✔ | - | - | - |
| SHPINX (Lin et al., 2023) | - | ✔ | ✔ | ✔ | - | ✔ | - | - | - | ✔ | ✔ | - | - | ✔ |
| LLaVA-Plus (Liu et al., 2023c) | ✔ | ✔ | ✔ | ✔ | ✔ | - | - | - | - | - | - | ✔ | - | - |
| LISA (Lai et al., 2024) | ✔ | - | ✔ | ✔ | ✔ | - | - | - | - | - | - | - | - | - |
| Osprey (Yuan et al., 2024) | ✔ | - | ✔ | ✔ | - | - | ✔ | - | - | - | - | - | - | - |
| GLaMM (Rasheed et al., 2024) | ✔ | - | ✔ | ✔ | ✔ | - | ✔ | - | - | ✔ | - | - | - | - |
| PixelLM (Xu et al., 2024) | ✔ | - | ✔ | ✔ | ✔ | - | - | - | - | ✔ | - | - | - | - |
| PSALM (Zhang et al., 2024b) | ✔ | - | ✔ | ✔ | ✔ | ✔ | ✔ | ✔ | ✔ | ✔ | - | ✔ | - | - |
| GroundHOG (Zhang et al., 2024a) | ✔ | - | ✔ | ✔ | ✔ | ✔ | ✔ | - | - | ✔ | - | - | - | - |
| F-LLM (Wu et al., 2024) | ✔ | - | ✔ | ✔ | ✔ | - | - | - | - | ✔ | - | - | - | - |
| VITRON (Fei et al., 2024) | - | ✔ | ✔ | ✔ | ✔ | - | ✔ | - | - | - | - | - | ✔ | - |
| VT-PLUG (Ours) | ✔ | ✔ | ✔ | ✔ | ✔ | ✔ | ✔ | ✔ | ✔ | ✔ | ✔ | ✔ | ✔ | ✔ |

To enhance the diversity of vision-language tasks, we propose **Visual-Task Instruction Following Dataset (VT-Instruct)**, a multimodal dataset specifically designed to support a wide range of

tasks, including Visual Understanding, Referring Expressions, Interactive Grounding (IG), Open-Vocabulary Identification, Grounded Conversation Generation (GCG), Keypoint Detection and Depth Estimation. VT-Instruct consists of more than 100 million high-quality multimodal dialogue samples, primarily derived from publicly available datasets such as LAION-5B (Schuhmann et al., 2022), SA-1B (Kirillov et al., 2023), COCO (Lin et al., 2014), GRIT (Peng et al., 2023), etc. Each sample is enhanced with thoughtfully crafted prompt templates with multimodal inputs (e.g. images, texts, points, boxes, scribbles and masks) to facilitate instruction following and diverse outputs (e.g. texts, boxes, keypoints, depth and masks) for different downstream tasks.

The contributions of this work can be summarized as follows:

- **VT-PLUG**: We propose a novel visual multi-task training framework that includes four meta-plugins designed to handle diverse visual content. These plugins can be combined to support various composite tasks and serve as the foundation for creating new visual plugins.

- **VD-CoT**: We propose a visual information generation method, Visual Decoding Chain-of-Thought (VD-CoT), for unified instruction tuning. VD-CoT provides VT-PLUG with learnable queries for visual unit decoding, along with essential auxiliary information, such as visual content descriptions and decoding unit types.

- **VT-Instruct**: We present a large-scale multi-task dataset containing 100 million multi-modal dialogue samples across 25 task types, which supports a comprehensive understanding and decoding of visual units across various degrees of prediction density.

- Quantitative experiments demonstrate that our VT-PLUG outperforms current MLLMs across multiple tasks. Specifically, VT-PLUG surpasses Osprey (Yuan et al., 2024) by 2.5 in CIDEr for Referring Expression Generation (REG), outperforms GLaMM (Rasheed et al., 2024) by 8.2% in Recall for the Grounded Conversation Generation (GCG), and exceeds PSALM (Zhang et al., 2024b) by 2.8% in $mAP_S$ for Open-Vocabulary Segmentation.

## 2 RELATED WORKS

Numerous studies have attempted to enhance the robust scene understanding capabilities of MLLMs, guiding models to achieve precise localization of identified objects. Pix2Seq (Chen et al., 2021) leverages the model's autoregressive generation capability to express bounding boxes and class labels as sequences of discrete tokens. Shikra (Chen et al., 2023b) constructs an appropriate visual supervision fine-tuning dataset, where the model needs to perform inductive analysis in the form of Chain-of-Thought (CoT) before answering complex questions, and subsequently outputs bounding boxes in text form to complete the visual grounding task.

LLaVa-Plus (Liu et al., 2023c) constructs an instruction-following dataset that includes a large number of samples for using task-specific models as tools. The model, after supervised fine-tuning, can leverage various task-specific models to accomplish tasks such as visual grounding and referring segmentation. LISA (Lai et al., 2024) adopts SAM (Kirillov et al., 2023) as the mask decoder, where MLLM generates learnable special tokens as prompts for SAM, producing fine-grained segmentation results. PSALM (Zhang et al., 2024b) divides the input for open-vocabulary segmentation tasks into instruction prompts, condition prompts, and discrete mask tokens, decoding the output mask tokens to obtain segmentation results aligned with the prompt content.

Unlike the existing research on visual fine-grained localization, our VT-PLUG is designed with four distinct meta-plugins that eliminate the need for additional task-specific models, ensuring overall consistency and accuracy.

## 3 UNIFIED INSTRUCTION TUNING FOR VISUAL UNIT DECODING

In general vision-language multimodal tasks, diverse user prompt inputs and visual unit outputs can extend the application of MLLMs to real-world scenarios. In Section 3.1, we introduce the **Visual Decoding Chain-of-Thought (VD-CoT)**, an instruction tunning approach designed to integrate various vision-language unit decoding tasks. In Section 3.2, we present **Visual-Task Instruction Following Dataset (VT-Instruct)**, a large-scale visual multi-task dataset that combines different visual prompts as inputs and visual units as outputs.

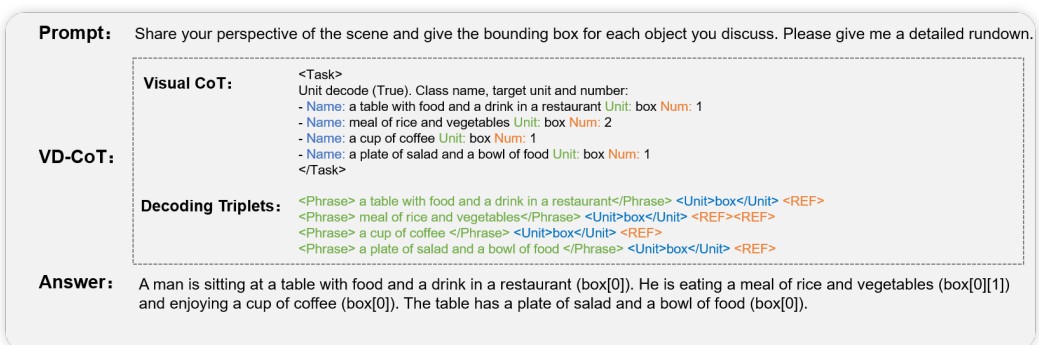

Figure 2: **An Example of VD-CoT Applied to the Grounded Conversation Generation (GCG) Task.** VD-CoT consists of two steps: Visual CoT for analyzing the visual content and Decoding Triplets for generating the decoding information triplet. The answer is generated synchronously with the triplet, and the special tokens have been simplified in the example.

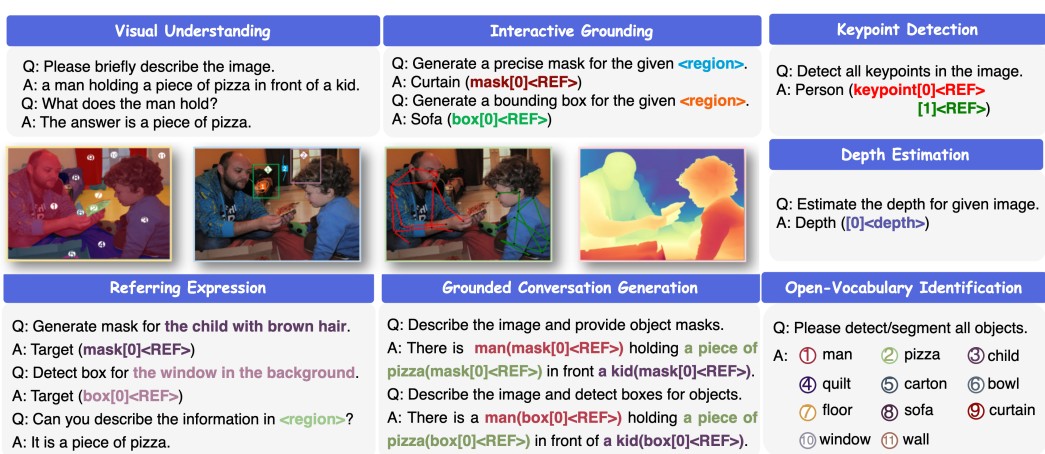

Figure 3: **Example of VT-Instruct Dataset by Using the Automated Data Construction Pipeline.** Our VT-Instruct dataset contains seven distinct downstream tasks, including Visual Understanding, Referring Expression, Interactive Grounding, Grounded Conversation Generation, Open-Vocabulary Identification and Depth Estimation.

## 3.1 VD-CoT

The decoding process of visual units requires essential information, including the description of visual entities, the type of unit, and the current decoding token. We add special tokens to the vocabulary of the MLLM to encode or mark the aforementioned content. The description of the visual object is denoted with `Phrase`, the decoding type is marked using `Unit`, and the token that requires further decoding is denoted with `<REF>`.

VD-CoT divides the answer generation process into two steps: visual CoT and decoding triplet. As shown in Figure 2, in visual CoT step, the model considers the visual entities to be decoded, the number of instances, and the type of decoding required for the current task. As for decoding triplet, it is generated simultaneously with the answer. For each `<REF>` token used for decoding, the model produces a `Phrase-Unit-<REF>` triplet. The answer content shown in the example omits the extra special tokens for better visualization.

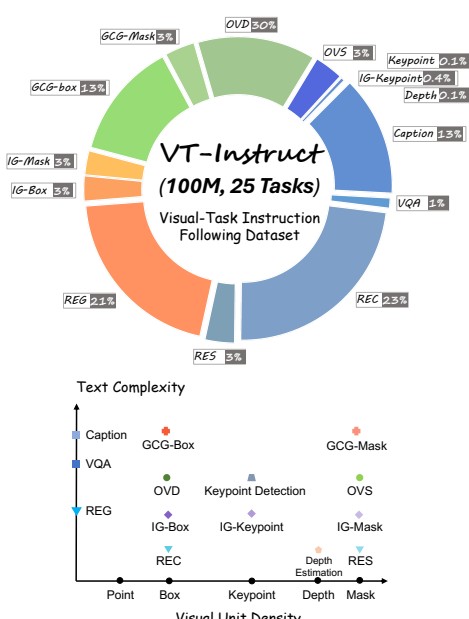

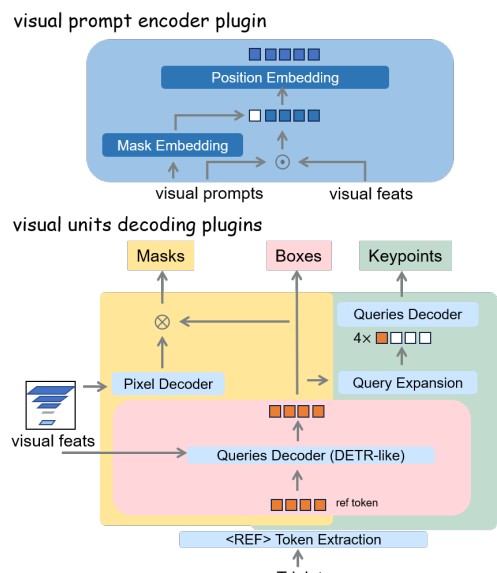

Figure 4: **Data Distribution Map.** VT-Instruct comprises four output units—box, keypoint, depth, and mask—paired with either low (phrases) or high (sentences) text complexity, with different visual prompts unified under the same task for clarity.

Figure 5: **Architecture of Visual Plugins.** Benefiting from the `Phrase-Unit-<REF>` triplet, where each `<REF>` token has a unique corresponding phrase and unit, thus ensuring the consistency of visual entities recognition and visual units decoding processes.

## 3.2 VT-INSTRUCT

**Multi-task Instruction Following Dataset.** We construct the VT-Instruct dataset, which comprises over 100 million dialogue samples featuring multimodal input-output pairs. These pairs encompass various combinations of output units, ranging from low to high visual density, including `Point`, `Box`, `Keypoint`, `Depth`, and `Mask`, combined with either low or high text complexity. VT-Instruct supports a wide range of tasks, facilitating both vision-language and dense prediction tasks, such as Visual Understanding, Referring Expression, Interactive Grounding, Open-Vocabulary Identification, Grounded Conversation Generation, Keypoint Detection and Depth Estimation (see Figure 4). Visual Understanding task includes Image Captioning and Visual Question Answering (VQA). Referring Expression tasks cover Referring Expression Comprehension (REC), Referring Expression Segmentation (RES), and Referring Expression Generation (REG). Interactive Grounding (IG) contains interactive detection (IG-box), segmentation (IG-mask) and keypoints generation(IG-keypoint). Open-Vocabulary Identification includes Open-Vocabulary Detection (OVD) and Open-Vocabulary Segmentation (OVS). Grounded Conversation Generation (GCG) could be divided into GCG-box and GCG-mask. The details of definition for each task will be presented in Appendix A.1.

**VT-Instruct Construction Pipeline.** For each downstream task, we **(i)** first construct a specific system instruction and **(ii)** generate over 150 task-specific prompt templates using GPT-4, randomly selecting them to construct user prompts, then **(iii)** we modify existing dataset annotations to construct a unified answering format following the rule of **VD-CoT** (Section 3.1), creating multi-turn conversations featuring a system-prompt-answer combination. Figure 3 illustrates an example of an image created using our automated pipeline, designed to support multiple downstream tasks.

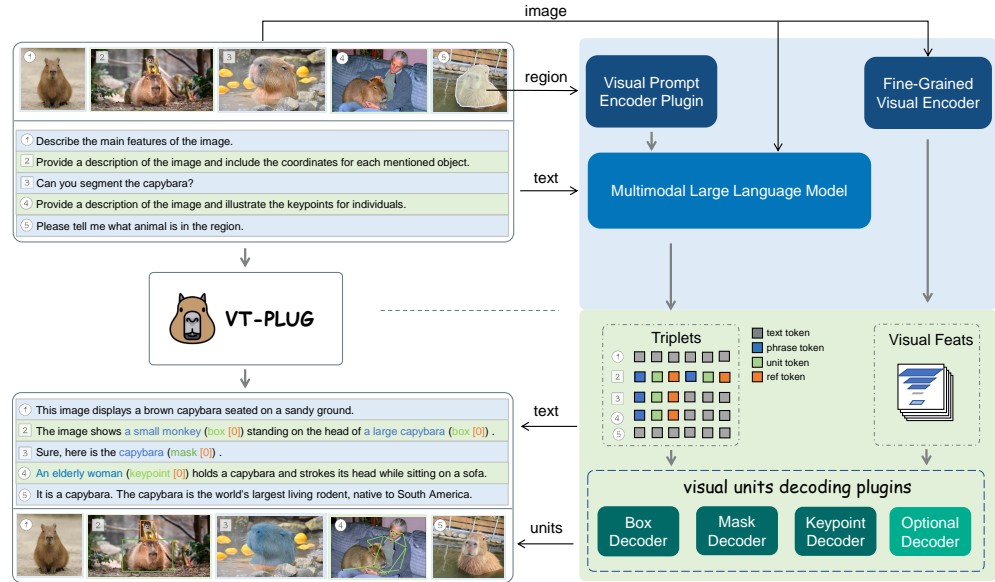

Figure 6: **The Framework of VT-PLUG.** The overall architecture consists of an MLLM, Conv-Encoder, VPT-Encoder, and Visual Units Decoders. The Conv-Encoder is responsible for generating multi-scale visual features, the VPT-Encoder encodes different forms of input, and the Visual Units Decoders flexibly support task-specific selections.

# 4 UNIFIED FRAMEWORK FOR VISUAL TASK PLUGINS INTEGRATION

## 4.1 PRELIMINARY

MLLMs often lack the capability to output visual units such as boxes, keypoints, and masks. To expand their applicability in real-world visual tasks, it is typically necessary to implement targeted designs for different visual tasks. Common decoding approaches for visual units can be categorized into three main types.

**Decode Visual Units as Sequence.** The most straightforward solution leverages the text generation capabilities of MLLM to produce visual localization results in textual format (Chen et al., 2023b; 2021). This approach does not require structural modifications to the MLLM. It simply necessitates the preparation of suitable supervised fine-tuning data to effectively generate the localization coordinates of visual targets. However, due to the constraints of textual output, their models struggle with dense prediction tasks such as keypoint detection, segmentation, and depth estimation.

**Decode Visual Units with Agent Tools.** Another approach involves using the MLLM as an agent to coordinate task-specific models, enabling accurate localization of visual targets (Liu et al., 2023c). In this case, MLLM outputs textual descriptions of recognized content and scheduling results, which can be utilized by downstream visual tools. However, since the final visual units is derived from the tool models, there may be a gap between the MLLM's understanding and the final output.

**Decode Visual Units with Learnable Queries.** Using the tokens output by the MLLM as learnable queries input into task-specific decoders is the most widely adopted visual decoding strategy (Rasheed et al., 2024; Lai et al., 2024). Directly decoding MLLM output tokens enables an end-to-end training process, allowing the visual decoder to share the visual fetures with the MLLM, thereby maintaining consistency at the feature level and achieving high accuracy and coherence. However, different types of decoding units (such as boxes, masks, and keypoints) require distinct visual information, resulting in significant variations in the data formats needed during supervised fine-tuning. Most research typically focuses on designing for a single decoding unit, making it challenging to integrate various decoding tasks into a unified instruction tuning framework.

## 4.2 VT-PLUG FRAMEWORK

Our VT-PLUG implements end-to-end unified training for multiple visual tasks, containing four meta-plugins to support the combination of various visual prompts and decode units. As shown in Figure 6, the VT-PLUG framework consists of four main components:

1. **Fundamental MLLM:** An Llava-like (Liu et al., 2023b) MLLM, with CLIP-ViT (Radford et al., 2021) as the visual encoder and Vicuna-7B (Zheng et al., 2023) as the LLM component.

2. **Fine-Grained Visual Encoder:** CLIP-ViT (Radford et al., 2021) focuses on encoding global image features, whereas visual units decoding tasks typically rely more heavily on local image features. To address this, we use CLIP-ConvNeXt (Cherti et al., 2023) for secondary image encoding, functioning as a visual feature pyramid encoder. In practice, we exclude the final layer of features encoded by CLIP-ConvNeXt and concatenate the remaining features with those from CLIP-ViT. By utilizing these two distinct visual encoders, we achieve an effective fusion of global and local image features representing.

3. **Visual Prompts Encoder Plugin:** We define user inputs such as points, boxes, scribbles and masks as Visual Prompts (VPT) and propose a novel VPT-Encoder plugin to encode these elements.

4. **Visual Units Decoder Plugins:** We propose three different decoding plugins to handle the decoding of boxes, keypoints, and masks.

As described in Section 3.1, when handling visual tasks, VT-PLUG performs a VD-CoT process based on the input prompts, analyzing the decoding content, decode unit, and decode target to form a `Phrase-Unit-<REF>` triplet. All `<REF>` tokens are extracted from the triplets, arranged in sequence, and used as learnable queries, which are then decoded by the visual decode plugins.

Figure 5 illustrates the architecture of our proposed visual plugins. The Visual Prompt Encoder plugin performs mask embedding for input regions. Unlike similar works that perform mask pooling on regions, our approach additionally incorporates position embedding to enhance the positional information of visual prompt features. The Visual Unit Decoding plugins adhere to the DETR framework (Carion et al., 2020). Since MLLMs can directly output classification results in the form of phrases in triplets, we eliminate the class predictor. Specifically, the box decoder is implemented as in DETR, while the mask decoder is implemented as in MaskFormer (Cheng et al., 2021). For the keypoint decoder, we develop a query expansion module after the box queries decoder. This module concatenates each query with a learnable vector initialized to zero and feeds the resulting representation into the subsequent queries decoder to predict the coordinates of the keypoints.

For the multi-target decoding process (i.e., a single `Phrase` corresponding to multiple `<REF>` instances), we employ the Group Hungarian Matcher (Carion et al., 2020). In the VT-PLUG setting, REF tokens are annotated in one-to-one correspondence with units, and we can achieve perfect matching by executing the Hungarian algorithm within each phrase group.

## 5 EXPERIMENTS

We conduct quantitative evaluations of our VT-PLUG across the following tasks details in Section 5.1: (i) Visual Understanding, (ii) Referring Expression, (iii) Interactive Grounding, (iv) Open-Vocabulary Identification, (v) Grounded Conversation Generation (GCG). Then, we perform ablation studies to evaluate the effectiveness of the key elements in our approach in Section 5.2. The training details of our VT-PLUG are presented in Appendix A.3

### 5.1 QUANTITATIVE RESULTS

**Visual Understanding.** We first present quantitative comparisons on zero-shot image captioning tasks using the prompt "`<image> Please describe the image in detail`" on the Flickr30k validation dataset (Plummer et al., 2015). For the VQA tasks, we employ the prompt "`Please take a look at the image <image> and promptly provide`

Table 2: Comparing VT-PLUG with other MLLMs on VQA and Image Captioning.

| Task | Datasets | VT-PLUG | Shikra | FM-80B | FM-9B | Kosmos-2 | Kosmos-1 | Flamingo-9B | Ferret-7B |
|---|---|---|---|---|---|---|---|---|---|
| VQA | VQAv2dev | 77.34 | 77.36 | 56.3 | 51.8 | 45.6 | 46.7 | 51.8 | - |
| | VQAv2std | 77.42 | 77.51 | - | - | - | - | - | - |
| | OK-VQA | **62.39** | 47.16 | 50.6 | 44.7 | - | 45.9 | - | - |
| Caption | Flickr30k | **85.25** | 73.9 | 67.2 | 61.5 | 66.7 | 65.2 | 61.5 | 74.8 |

Table 3: Object hallucination benchmark in three POPE (Li et al., 2023) evaluation settings.

| Sampling | Metrics | VT-PLUG | Osprey | Ferret | Shikra | LLaVA-1.5 | Instruct-BLIP | MiniGPT4 | MM-GPT | mPLUG-Owl |
|---|---|---|---|---|---|---|---|---|---|---|
| Random | Accuracy | 87.63 | 89.47 | 90.24 | 86.90 | 88.73 | 88.57 | 79.67 | 50.10 | 53.97 |
| | Precision | **97.98** | 93.40 | 97.72 | 94.40 | 88.89 | 84.09 | 78.24 | 50.05 | 52.07 |
| | Recall | 77.60 | 84.93 | 83.00 | 79.26 | 88.53 | 95.13 | 82.20 | 100.00 | 99.60 |
| | F1 Score | 86.61 | 88.97 | 89.76 | 86.19 | 88.71 | 89.27 | 80.17 | 66.71 | 68.39 |
| | Yes(%) | 40.82 | 45.47 | 43.78 | 43.26 | 49.80 | 56.57 | 52.53 | 99.90 | 95.63 |
| Popular | Accuracy | 86.27 | 87.83 | 84.90 | 83.97 | 85.83 | 82.77 | 69.73 | 50.00 | 50.90 |
| | Precision | **93.94** | 89.94 | 88.24 | 87.55 | 83.91 | 76.27 | 65.86 | 50.00 | 50.46 |
| | Recall | 77.53 | 85.20 | 80.53 | 79.20 | 88.67 | 95.13 | 81.93 | 100.00 | 99.40 |
| | F1 Score | 84.95 | 87.50 | 84.21 | 83.16 | 86.22 | 84.66 | 73.02 | 66.67 | 66.94 |
| | Yes(%) | 41.27 | 47.37 | 45.63 | 45.23 | 52.83 | 62.37 | 62.20 | 100.00 | 98.57 |
| Adversarial | Accuracy | 84.97 | 85.33 | 82.36 | 83.10 | 72.10 | 65.17 | 79.20 | 50.00 | 50.67 |
| | Precision | **90.75** | 85.43 | 83.60 | 85.60 | 74.69 | 65.13 | 61.19 | 50.00 | 50.34 |
| | Recall | 77.87 | 85.20 | 80.53 | 59.60 | 88.34 | 95.13 | 82.93 | 100.00 | 90.33 |
| | F1 Score | 83.82 | 85.31 | 82.00 | 82.49 | 80.94 | 77.32 | 70.42 | 66.67 | 66.82 |
| | Yes(%) | 42.90 | 49.87 | 48.18 | 46.50 | 59.14 | 73.03 | 67.77 | 100.00 | 98.67 |

an answer for <question>" on the VQAv2-dev, VQAv2-std (Antol et al., 2015), and OK-VQA (Marino et al., 2019) test datasets. We report overall accuracy for the VQA tasks and the CIDEr score for the image captioning task. The comparison results are summarized in Table 2, where our VT-PLUG achieves the best performance on the image captioning task with a CIDEr score of 85.25 and on the OK-VQA test dataset with 62.39% accuracy, while demonstrating competitive performance on the VQAv2 test dataset, comparable to Shikra (Chen et al., 2023b). Additionally, we employ the POPE benchmark (Li et al., 2023) to evaluate hallucination performance in VT-PLUG in Table 3. In each case, VT-PLUG achieves the highest precision, outperforming other MLLMs.

Table 4: Evaluation results for referring expression tasks, including RES and REC. "w/o pretrained" indicates whether the segmentation model used a pretrained backbone for the RES task, while ✔ indicates that the model was trained from scratch. ZS denotes that the result was obtained in a zero-shot setting, while FT indicates the model was finetuned on the RefCOCO training dataset.

| Task | Model | w/o pretrained | RefCOCO | | | RefCOCO+ | | | RefCOCOg | |
|---|---|---|---|---|---|---|---|---|---|---|
| | | | Test-A | Test-B | Val | Test-A | Test-B | Val | Test | Val |
| RES (cIOU) | MCN | ✘ | 64.2 | 59.7 | 62.4 | 55.0 | 44.7 | 50.6 | 49.4 | 49.2 |
| | VLT | ✘ | 70.5 | 65.2 | 67.5 | 61.0 | 50.1 | 56.3 | 57.7 | 55.0 |
| | CRIS | ✘ | 73.2 | 66.1 | 70.5 | 68.1 | 53.7 | 62.3 | 60.4 | 59.9 |
| | LAVT | ✘ | 75.8 | 68.8 | 72.7 | 68.4 | 55.1 | 62.1 | 62.1 | 61.2 |
| | RELA | ✘ | 76.5 | 70.2 | 73.8 | 71.0 | 57.7 | 66.0 | 66.0 | 65.0 |
| | X-Decoder | ✘ | - | - | - | - | - | - | - | 64.6 |
| | SEEM | ✘ | - | - | - | - | - | - | - | 65.7 |
| | LISA | ✘ | 76.5 | 71.1 | 74.1 | 67.4 | 56.5 | 62.4 | 68.5 | 66.4 |
| | VT-PLUG(ZS) | ✔ | 71.6 | 57.5 | 65.6 | 63.8 | 48.1 | 59.3 | 62.1 | 58.4 |
| | VT-PLUG(FT) | ✔ | 73.4 | 63.9 | 69.0 | 70.8 | 56.2 | 63.2 | 65.8 | 65.0 |
| REC (IOU>0.5) | OFA-L | - | 83.7 | 76.4 | 76.4 | 76.0 | 61.8 | 68.3 | 67.6 | 80.0 |
| | MAttNet | - | 80.4 | 69.3 | 80.0 | 70.3 | 56.0 | 64.9 | 67.0 | 76.4 |
| | Kosmos-2 | - | 57.4 | 47.3 | 52.3 | 50.7 | 42.2 | 45.5 | 61.7 | 60.6 |
| | Shikra | - | 90.6 | 80.2 | 87.0 | 87.4 | 72.1 | 81.6 | 82.2 | 82.3 |
| | Ferret | - | 91.4 | 82.5 | 87.5 | 87.4 | 73.1 | 80.8 | 84.8 | 83.9 |
| | NeXT-Chat | - | 90.0 | 77.9 | 85.5 | 84.5 | 68.0 | 77.2 | 79.8 | 80.1 |
| | VT-PLUG(ZS) | - | 90.7 | 78.5 | 85.2 | 86.0 | 67.2 | 77.0 | 80.0 | 80.2 |
| | VT-PLUG(FT) | - | **92.5** | 82.3 | **88.3** | **88.3** | **73.7** | **81.7** | 83.0 | 83.1 |

**Referring Expression.** For the Referring Expression Segmentation (RES) task, we evaluate VT-PLUG on the RefCOCO, RefCOCO+, and RefCOCOg test and validation datasets by calculating the cumulative IOU (cIOU) as proposed by Liu et al. (2023a), using the prompt "`Provide a segmentation mask for <referring expression> in the picture <image>.`" Our VT-PLUG, trained from scratch, achieves results in both zero-shot and fine-tuned settings that are comparable to recent methods like LISA (Lai et al., 2024), which utilized pretrained backbones such as SAM (see Table 4). For the Referring Expression Comprehension (REC) task, we use the prompt "`What are the coordinates of <referring expression> in the image<image>?`" and compare our VT-PLUG with current MLLMs capable of generating referring boxes based on specific prompts in both zero-shot and fine-tuned settings. The metric used for REC evaluation is ACC@0.5. As shown in Table 4, VT-PLUG demonstrates superior performance in the REC task compared to other MLLMs. We evaluate Referring Expression Generation (REG) using the prompt, "`For the given image <image>, can you provide a unique description of the area <mask>?`" on the RefCOCOg test dataset (Kazemzadeh et al., 2014). The evaluation metrics applied are Meteor and CIDEr, with the results presented in Table 5. Our VT-PLUG demonstrates improved performance compared to GLaMM (Rasheed et al., 2024) and Osprey (Yuan et al., 2024), while also showing robust zero-shot capabilities.

Table 5: REG Evaluation on RefCOCOg.

| Model | Type | Meteor | CIDEr |
|-------|------|--------|-------|
| GRIT | Box | 15.2 | 71.6 |
| Kosmos-2 | Box | 14.1 | 62.3 |
| GLaMM(FT) | Box | 16.2 | 105.0 |
| Osprey(FT) | Mask | 16.6 | 108.3 |
| VT-PLUG(ZS) | Mask | 15.8 | 98.1 |
| VT-PLUG(FT) | Mask | **16.9** | **110.8** |

Table 6: Evaluation on COCO-interactive.

| Model | w/o pretrained | Scribble | Box | Mask |
|-------|----------------|----------|-----|------|
| SAM-B | ✔ | - | 68.7 | - |
| SAM-L | ✔ | - | 71.6 | - |
| SEEM-B | ✗ | 44.0 | 42.1 | 65.0 |
| PSALM | ✗ | 80.0 | 80.9 | 82.4 |
| VT-PLUG | ✔ | 60.2 | 73.7 | 77.5 |

**Interactive Grounding.** For this task, we evaluate using the prompt, "`Please generate a mask based on the region <region> in the image <image>.`" where `<region>` is replaced with visual prompts such as scribbles, boxes, or masks. The results presented in Table 6 indicate that our VT-PLUG outperforms both SAM (Kirillov et al., 2023) and SEEM-B (Zou et al., 2024) across the scribble, box, and mask settings, achieving performance comparable to PSALM, which utilizes pretrained Swin-T and Mask2Former weights in these configurations.

Table 7: VT-PLUG performance on Grounding Conversation Generation (GCG) task.

| Model | Dataset | Type | w/o SAM | Val | | | | | Test | | | | |
|-------|---------|------|---------|-------|--------|------|------|--------|-------|--------|------|------|--------|
| | | | | CIDEr | Meteor | AP50 | mIOU | Recall | CIDEr | Meteor | AP50 | mIOU | Recall |
| BuboGPT | | Mask | ✗ | 3.6 | 17.2 | 19.1 | 54.0 | 29.4 | 3.5 | 17.1 | 17.3 | 54.1 | 27.0 |
| Kosmos-2 | GranD$_f$ | Mask | ✗ | 27.6 | 16.1 | 17.1 | 55.6 | 28.3 | 27.2 | 15.8 | 17.2 | 56.8 | 29.0 |
| LISA | | Mask | ✗ | 33.9 | 13.0 | 25.2 | 62.0 | 36.3 | 32.2 | 12.9 | 24.8 | 61.7 | 35.5 |
| GLaMM | | Mask | ✗ | 47.2 | 16.2 | 30.8 | 66.3 | 41.8 | 37.9 | 14.6 | 27.2 | 64.6 | 38.0 |
| VT-PLUG | | Mask | ✔ | **56.9** | **18.4** | 26.2 | 57.9 | **50.0** | **53.2** | **21.7** | 27.7 | **56.6** | **45.3** |
| VT-PLUG | Flickr30k | Box | - | - | - | - | - | - | 82.0 | 26.0 | 35.4 | 66.1 | 47.7 |

**Grounded Conversation Generation (GCG).** The Grounded Conversation Generation (GCG) task consists of two components: GCG-mask and GCG-box. For the GCG-mask task, we further finetune our VT-PLUG on the GranD$_f$ training dataset and evaluate its performance on the GranD$_f$ validation and test splits, following the process outlined by Rasheed et al. (2024). We utilize the prompt, "`Describe the setting of the image <image> and offer masks for each visible object.`" for the GCG-mask evaluation. The results presented in Table 7 demonstrate that our VT-PLUG outperforms current baseline methods, such as GLaMM, across metrics including CIDEr, Meteor, AP50, and Recall. Additionally, we assess the GCG-box task using the Flickr30k test set with the prompt, "`Please describe the image <image> and detect relevant bounding boxes.`" Due to the lack of available MLLMs for the GCG-box task, we only report our zero-shot performance on this dataset.

Table 8: Evaluation on open-vocabulary tasks.

| Model | Type | Ade20k | COCO |
|---|---|---|---|
| MaskCLIP | SEG | 6.0 | - |
| ODISE | SEG | 14.4 | - |
| SAN | SEG | 10.6 | - |
| PSALM | SEG | 9.0 | - |
| PSALM+LVIS | SEG | 13.9 | - |
| VT-PLUG (mAP$_S$) | DET/SEG | **16.7** | **26.7** |

Table 9: Ablation study on group matcher.

| Visual Encoder | Group Matcher | cIoU |
|---|---|---|
| ConvNeXt + ViT | ✘ | 61.47 |
| ConvNeXt + ViT | ✔ | **62.49** |

Table 10: Comparison across visual encoder. [0,1,2,3] means we choose all four feature layers from CLIP-ConvNeXt model, -2 means we only choose CLIP-ViT second last layer, [0,1,2,4] means we concatenate the first three feature layers from CLIP-ConvNeXt and the output feature map from CLIP-ViT.

| Visual Encoder | Size | Feature Dimension | cIoU |
|---|---|---|---|
| ConvNeXt | 320 | [0,1,2,3] | 41.94 |
| ConvNeXt | 336 | [0,1,2,3] | 46.08 |
| ViT | 336 | -2 | 60.44 |
| ConvNeXt + ViT | 320 | [0,1,2,4] | 60.02 |
| ConvNeXt + ViT | 512 | [0,1,2,4] | **61.83** |

**Open-Vocabulary Identification** Our VT-PLUG not only excels in performing GCG tasks, similar to current MLLMs (Rasheed et al., 2024; Chen et al., 2023b), but also demonstrates proficiency in open-vocabulary identification tasks, including open-vocabulary segmentation and detection with a simple prompt template: "`Please detect bounding boxes (segment objects) in the image<image>`." We calculate mAP$_S$ (detailed in Appendix A.2). For the open-vocabulary segmentation task, we evaluate VT-PLUG on the ADE20k test dataset, and for the open-vocabulary detection task, we assess its performance on the COCO2017 validation dataset. The results for both tasks are presented in Table 8. Notably, VT-PLUG achieves strong performance without any specialized design, outperforming other MLLMs (e.g., PSALM) and specialist models (e.g., SAN). Additionally, unlike other MLLMs, VT-PLUG also demonstrates the capability to perform open-vocabulary object detection.

## 5.2 ABLATION STUDY

To evaluate the effectiveness of the core components of our framework, we conduct the following ablation studies.

**Choose of Group Matchers.** To validate the effectiveness of our Group Hungarian Matcher, we perform an ablation study on its usage in the mask decoder for the RES task, using the RefCOCOg test dataset and cIoU as the evaluation metric. As shown in Table 9, applying the Group Hungarian Matcher for loss computation yields a significantly better performance compared to configurations without it, demonstrating its substantial impact on improving the overall accuracy.

**Different Configuration of Visual Encoders.** To investigate the effect of different configurations of CLIP vision encoders, including CLIP-ConvNeXt and CLIP-ViT, along with variations in image size and feature selection layers, we conduct experiments on the RES task using the RefCOCOg test dataset. As shown in Table 10, VT-PLUG achieves the highest performance when concatenating the CLIP-ConvNeXt and CLIP-ViT encoders with the setting of image size as 512×512.

## 6 LIMITATIONS AND CONCLUSION

In conclusion, this paper introduces a powerful and flexible visual multi-task learning framework, alongside the construction of a large-scale vision-language multimodal instruction-tuning dataset. This work effectively expands the applicability of MLLMs in real-world scenarios, and extensive experiments validate its effectiveness. However, the focus of this work is limited to the problem of visual units decoding, and it cannot yet effectively handle widely-used tasks such as image editing and video understanding. Consequently, this work should be regarded as a foundational baseline for visual units decoding.

## 7 ETHICS STATEMENT

Our research fully adheres to the ICLR Code of Ethics, ensuring ethical standards are maintained throughout the whole study.

## 8 REPRODUCIBILITY STATEMENT

To ensure the reproducibility of our research, we have provided comprehensive implementation details, including data construction, model architecture and hyperparameter settings. Additionally, all datasets and data processing steps are fully documented in the supplementary materials. We will also release the complete source code and instructions for reproducing our results.

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

# A Appendix

## A.1 VT-Instruct Construction

Table 11: Data statistics of VT-Instruct and actual use of dataset in the training process.

| Task | | Original Dataset | Construction Number | Actual Use |
|---|---|---|---|---|
| Visual Understanding | Caption | COCO, GranD, GRIT | 15,980,000 | 780,000 |
| | VQA | VQAv2, LLaVA-Instruct | 1,310,000 | 1,310,000 |
| Referring Expression | REC | RefCOCO, RefCOCO+, RefCOCOg, GranD, GRIT | 22,880,000 | 880,000 |
| | RES | RefCOCO, RefCOCO+, RefCOCOg, GranD | 3,880,000 | 680,000 |
| | REG | RefCOCO, RefCOCO+, RefCOCOg, GranD, GRIT, COCO-Interactive, Osprey, Visual Genome, Visual7W | 22,750,000 | 1,200,000 |
| Interactive Grounding | IG-Box | COCO-Interactive | 3,200,000 | 120,000 |
| | IG-Mask | COCO-Interactive | 3,200,000 | 120,000 |
| | IG-Keypoint | COCO | 500,000 | 140,000 |
| Grounded Conversation Generation | GCG-box | GRIT, GranD, Flickr30k-Entities | 15,630,000 | 540,000 |
| | GCG-mask | GranD, LLaVA-Grounding, PNG, OpenPSG | 4,000,000 | 450,000 |
| Open-Vocabulary Identification | OVD | GranD, GRIT, COCO-REM | 15,770,000 | 600,000 |
| | OVS | GranD, COCO-REM, ADE20k, Cityscapes | 3,795,000 | 600,000 |
| Keypoint Detection | | COCO | 140,000 | 140,000 |
| Depth Estimation | | Kitti, HRWSI, NYU | 150,000 | - |

**Definition of Each Downstream Task** The Visual Understanding task includes Image Captioning and Visual Question Answering (VQA), involving image-text inputs and text-only outputs. Referring Expression tasks cover Referring Expression Comprehension (REC), Referring Expression Segmentation (RES), and Referring Expression Generation (REG). While REC and RES require models to predict bounding boxes or masks in response to a query about a specific region in an image, REG involves generating descriptive text from visual inputs like points, boxes, scribbles, or masks. Interactive Grounding (IG) enables users to provide prompts via both text and interactive inputs (e.g., points, boxes, masks), allowing MLLMs to interpret and generate corresponding outputs. Open-Vocabulary Identification focuses on localizing and segmenting objects from descriptive text, even if the object categories were not part of the training data. Grounded Conversation Generation (GCG) produces natural language responses interwoven with bounding boxes or masks, with the GCG task further divided into GCG-box (bounding box outputs) and GCG-mask (mask outputs).

**Dataset Construction Details** For each task, we select a unique prompt-unit pair to develop task-specific instructions. For example, visual understanding task encompasses Image Captioning and Visual Question Answering (VQA), with image-text inputs and pure text outputs. To facilitate MLLMs in comprehending image-level information and addressing diverse questions, we construct conversations for visual understanding tasks using our proposed pipeline with the COCO (Lin et al., 2014), GranD (Rasheed et al., 2024), GRIT (Peng et al., 2023), VQAv2 (Antol et al., 2015), and LLaVA-instruct (Liu et al., 2023b) datasets, which collectively comprise over 15 million image-text pairs featuring multi-turn conversations. Referring expression tasks include Referring Expression Comprehension (REC), Referring Expression Segmentation (RES), and Referring Expression Generation (REG). The REC and RES tasks require the model to respond to a question or description regarding a specific area in an image, predicting bounding boxes or masks. In contrast, the REG task involves inputs such as points, boxes, scribbles, and masks, with the model expected to generate a descriptive response based on the visual prompts. We construct conversations for referring expression task from refCOCO (Kazemzadeh et al., 2014), refCOCO+ (Kazemzadeh et al., 2014), refCOCOg (Kazemzadeh et al., 2014), GranD (Rasheed et al., 2024), GRIT (Lin et al., 2014), Osprey (Yuan et al., 2024), Visual Genome (Krishna et al., 2017) datasets with more than 22 million samples. Interactive grounding allows users to provide prompts through both text and interactive elements, such as points, boxes, masks, or scribbles, enabling MLLMs to interpret these inputs and generate corresponding outputs, including bounding boxes or masks. We constructed interactive grounding samples using the COCO-interactive (Zhang et al., 2024b) dataset , which contains over 64 million examples. The open-vocabulary identification task focuses on localizing and segmenting

objects in an image based on descriptive text prompts, even if the specific object categories were not included in the model's training data. To equip VT-PLUG with zero-shot capabilities for object detection and segmentation—similar to traditional open-vocabulary detection models (e.g., YOLO-World (Cheng et al., 2024)) and segmentation models (e.g., SAM (Kirillov et al., 2023)) — we designed a multimodal conversation system using bounding boxes and masks annotations from the GRIT (Peng et al., 2023), GranD (Rasheed et al., 2024), COCO-REM (Singh et al., 2024), ADE20k (Zhou et al., 2017), and Cityscapes (Cordts et al., 2016) datasets, resulting in a corpus of over 20 million examples. Grounded conversation generation (GCG) aims to produce natural language responses interwoven with bounding boxes or object segmentation masks. The GCG task is divided into GCG-box, which outputs bounding boxes, and GCG-mask, which outputs masks. We developed these tasks using datasets that include captions and phrases associated with bounding box or mask annotations, such as Flickr30k-entities (Plummer et al., 2015), GranD (Rasheed et al., 2024), GRIT (Peng et al., 2023), LLaVA-grounding (Zhang et al., 2023b), OpenPSG (Zhou et al., 2024), and PNG (González et al., 2021), collectively comprising over 18 million annotations.

## A.2 AP Similarity ($AP_S$)

Instead of the calculating mAP as our evaluation metric for Open-Vocabulary Identification tasks, we propose a new metric called mAP Similarity ($AP_S$) to evaluate our VT-PLUG performance. For traditional open-vocabulary models, they typically predict classes with a logit score by their classification head. However, instead of applying a classification head for each task, our VT-PLUG leverages a large language model (LLM) to predict classes without generating any class logits. We therefore compute the similarity score between VT-PLUG's class predictions and all ground truth class names. We then assign the class label based on the highest similarity, using this similarity score in place of the traditional confidence score.

For the implementation of $AP_S$, we define the phrases predicted by the LLM as $p_i \in p_1, p_2, p_3, \ldots, p_k$, where $k$ denotes the number of LLM predictions. The ground truth classes are denoted as $c_i \in c_1, c_2, c_3, \ldots, c_n$, where $n$ is the total number of ground truth classes for the dataset. We first use the CLIP-Large-14-336 model to compute the text embeddings $e$, as shown in Equation (1). Next, we compute the cosine similarity score between each $p_i$ and $c_i$ as in Equation (2). The class of our predicted phrase is assigned based on the maximum similarity score and its corresponding index, which also serves as the logit score for the prediction.

$$e_{p_i} = \text{CLIP}(p_i), e_{c_i} = \text{CLIP}(c_i). \tag{1}$$

$$s_{max}, id_{max} = \max\left(\text{Cosine\_Similarity}(e_{p_i}, e_{c_i})\right). \tag{2}$$

## A.3 Training Details

The training process of VT-PLUG is conducted in three stages, during which both CLIP-ViT and CLIP-ConvNeXt are frozen, with no parameter updates. We use eight NVIDIA A800-80GB GPUs in all of our training processes and pick Vicuna-7B as our LLM, CLIP-large-14-336 and CLIP-ConvNeXt-512 as our visual encoder. In the first stage, VT-PLUG adopts the same setting as Shikra, freezing all model parameters except for the projector, aiming to achieve alignment of multimodal data, we train the first stage for about 2 days with setting the lr to $1e - 5$. In the second stage, VT-PLUG is trained using the VT-Instruct data that we constructed as shown in Section 3.2, updating parameters for all modules except the keypoint decoder. The goal of this stage is to train the LLM and various visual plugins using large-scale data, while the keypoint decoder is excluded from training due to its strong correlation with the box decoder. In the third stage, VT-PLUG continues training on the VT-Instruct dataset, updating all modules, with the keypoint decoder initialized with the weights of the box decoder from the second stage. We set the lr to $2e - 6$ in the second and third stages. It took about 7 days to complete the whole training process.

A.4 MULTI-TASKS RESULTS

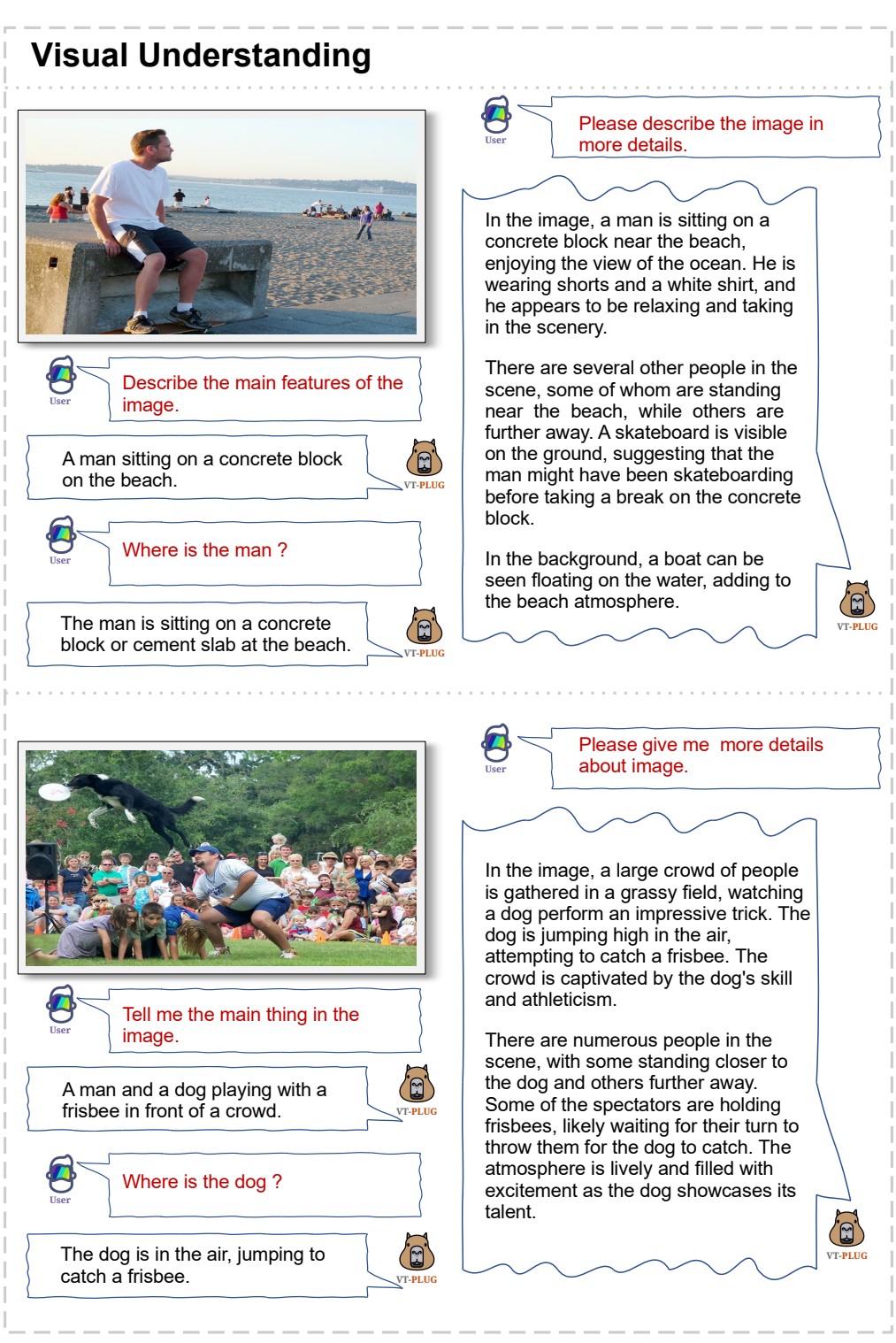

Figure 7: **The Visual Understanding Results of VT-PLUG.**

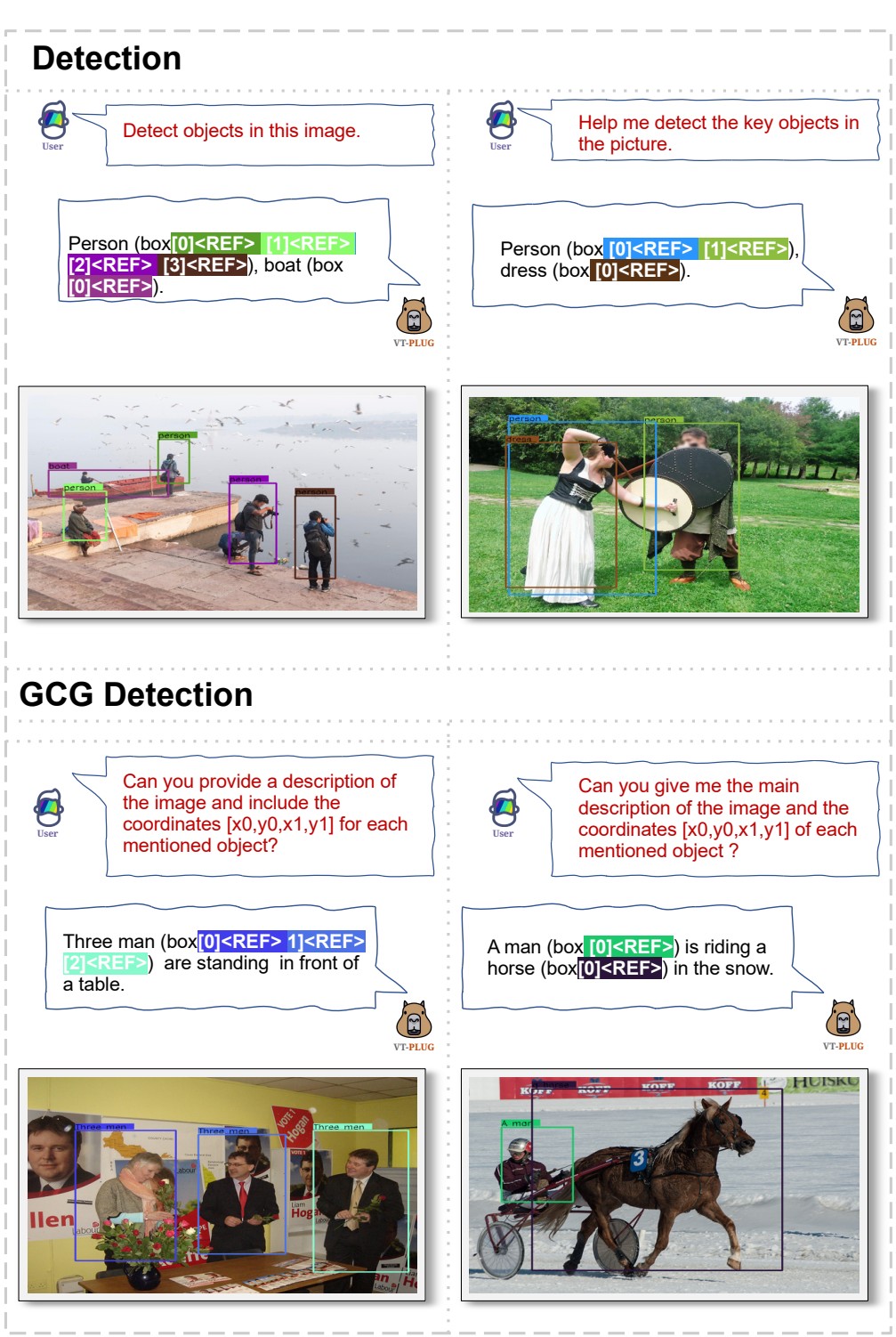

Figure 8: **The Detection Results of VT-PLUG.**

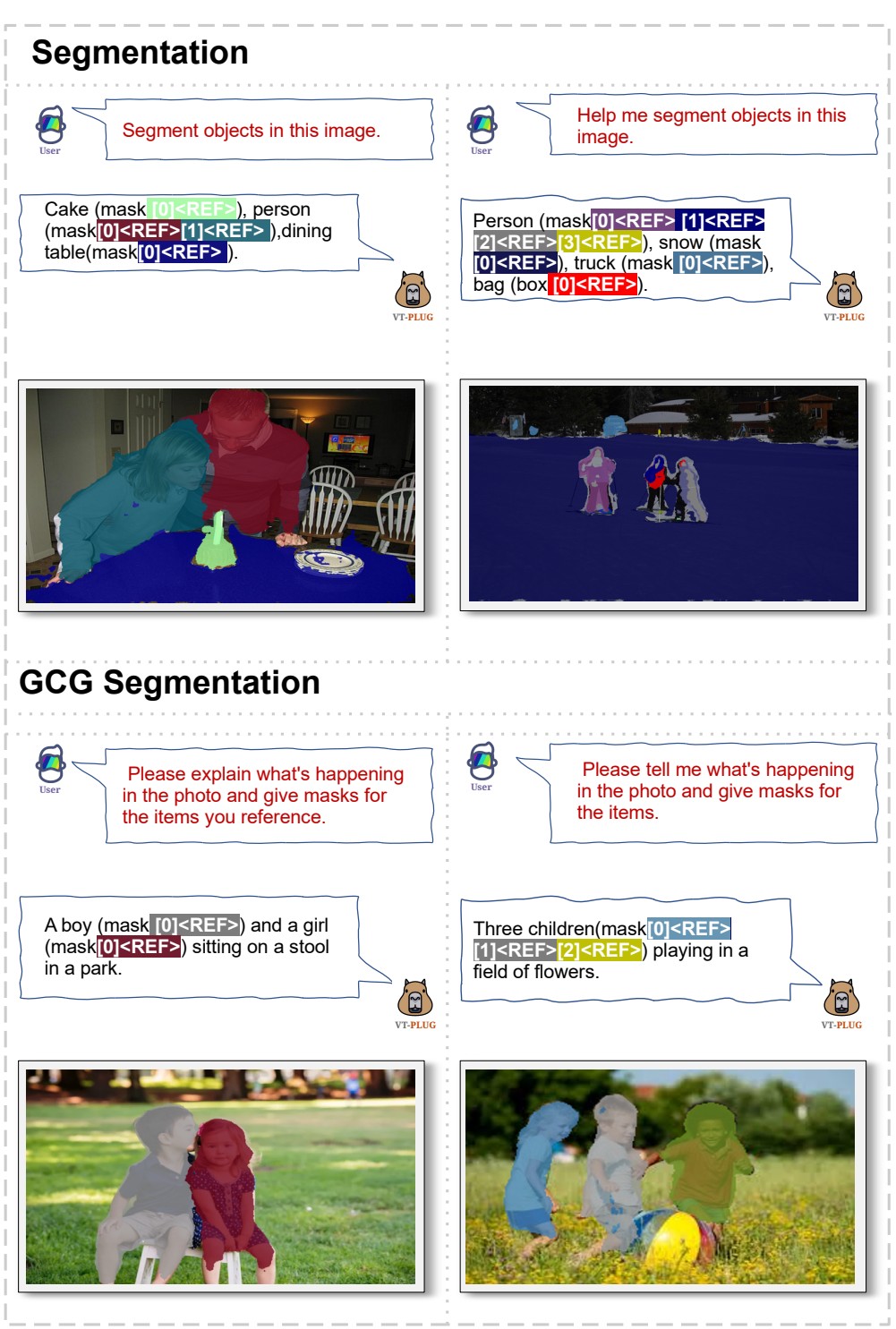

Figure 9: **The Segmentation Results of VT-PLUG.**

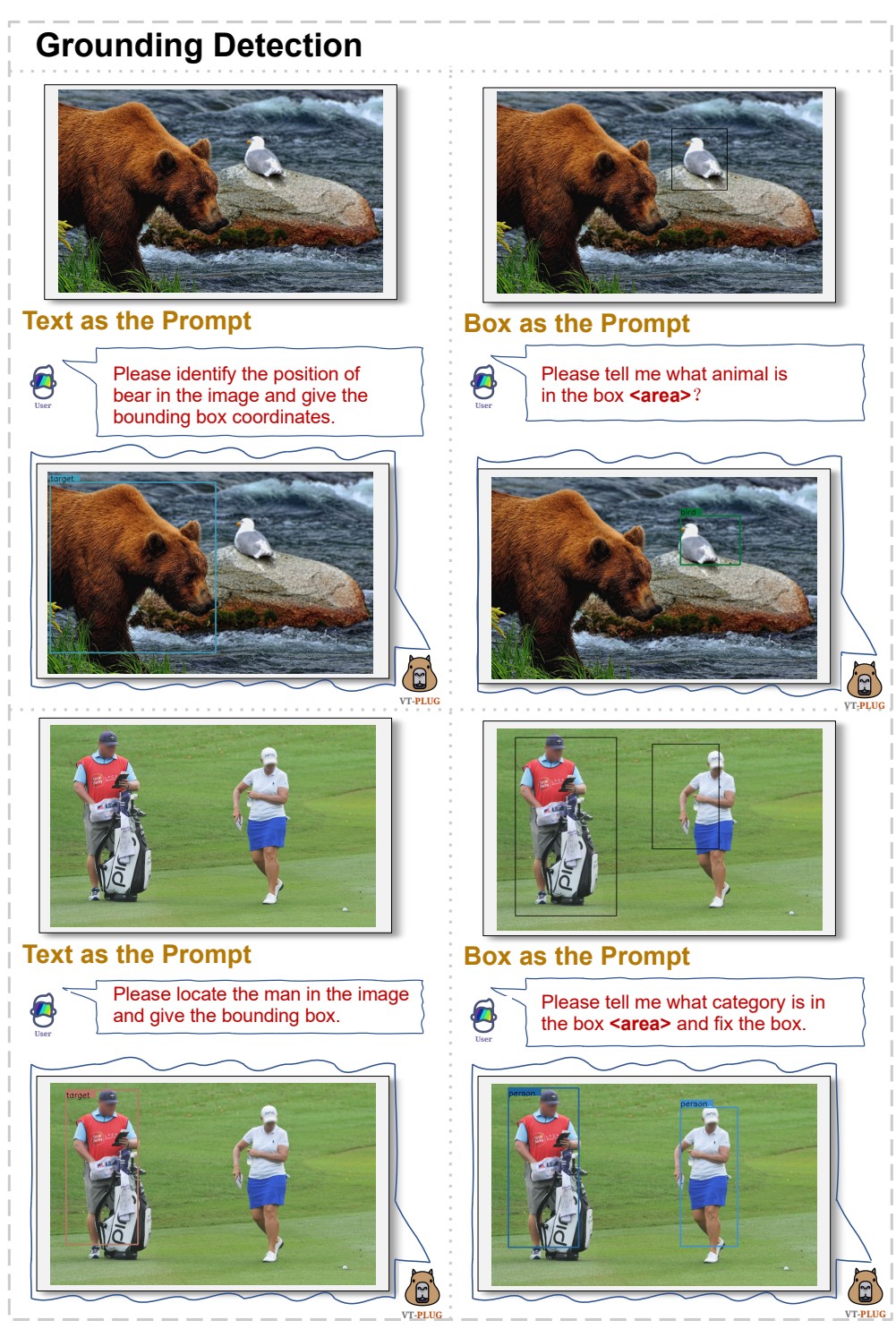

Figure 10: **The Grounding Detection Results of VT-PLUG.**

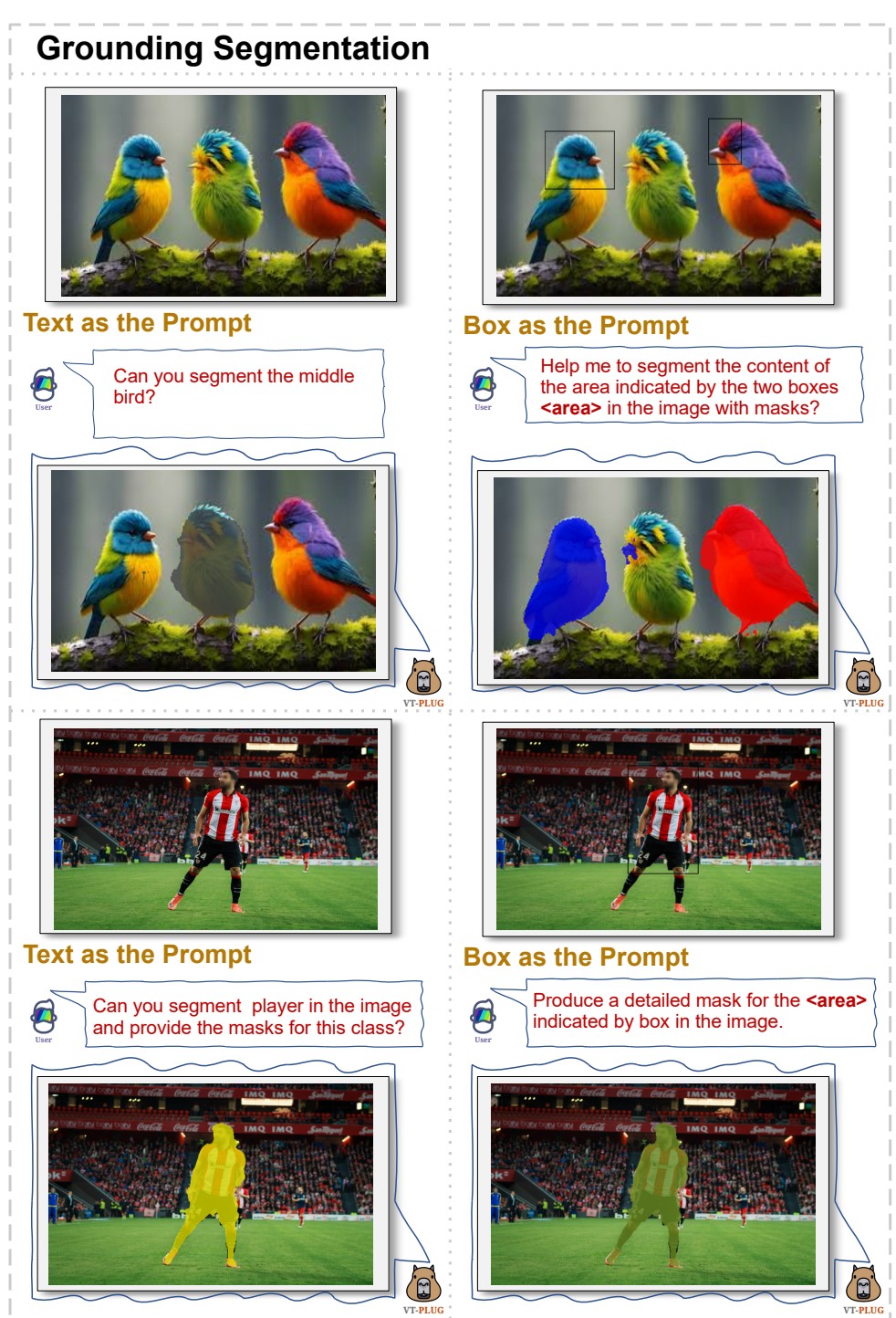

Figure 11: **The Grounding Segmentation Results of VT-PLUG.**

