# OpenReview forum: "VT-PLUG: Integrating Visual Task Plugins with Unified Instruction Tuning"
_ICLR.cc/2025/Conference — ICLR 2025 Conference Withdrawn Submission_

### Official Review · Reviewer_iSEr · 2024-11-01

**Soundness:** 2
**Presentation:** 3
**Contribution:** 2
**Rating:** 3
**Confidence:** 4

**Summary:**

This paper presents the "VT-PLUG" framework, which integrates multiple visual task plugins through unified instruction tuning to address a variety of vision-language tasks.

**Strengths:**

1. The paper constructs a large-scale and diverse multi-task dataset, VT-Instruct, which includes various visual prompts (e.g., points, boxes, scribbles, masks) and visual unit outputs (e.g., boxes, key points, depth, masks). This dataset significantly enhances the diversity of training data for multimodal models, providing a valuable resource for future multi-task research.

2. The paper conducts quantitative evaluations of VT-PLUG on various tasks, including visual understanding, referring expressions, interactive grounding, open vocabulary identification, and grounded conversation generation.

**Weaknesses:**

I have some concerns as follows:

1. The modular components of VT-PLUG largely overlap with existing multimodal model plugin integration frameworks. In addition, the proposed VD-CoT mechanism fails to clearly demonstrate its unique advantages over existing chain-of-thought (CoT) and visual decoding methods, giving the impression of merely borrowing the CoT concept.

2. Although the paper claims high performance across various tasks, the results do not significantly surpass state-of-the-art models on tasks such as Visual Question Answering (VQA) and image captioning. Additionally, it lacks a comparative analysis with advanced vision-language models, making it difficult to justify the added complexity of the proposed method without substantial performance improvements.

3. The framework's reliance on separate plugins for each visual task increases system complexity, potentially limiting its scalability and applicability in real-world scenarios. This integration approach may incur high computational costs, which the paper does not adequately address.

4. Notable comparable methods, such as the VisionLLM v2 series, which shares similarities in task and model design, are not discussed. Additionally, pioneering works introducing visual plugins, such as Visual ChatGPT, HuggingGPT, and MM-REACT, are also absent from the discussion.

Overall, this paper does not offer a truly novel and impactful solution. Due to the overlap with existing methods, limited empirical improvements, and the lack of clarity in articulating technical contributions, my rating is "reject".

**Questions:**

See weaknesses

---

### Official Review · Reviewer_ZpUw · 2024-11-03

**Soundness:** 2
**Presentation:** 2
**Contribution:** 2
**Rating:** 3
**Confidence:** 4

**Summary:**

This work proposes VT-PLUG, a unified multimodal large language model (MLLM), to address various vision-language tasks. The model is able to take user inputs provided by any combinations of text, images, and visual prompts such as boxes and masks, and decode multimodal contents including text, boxes, masks, and keypoints. To learn this model, Visual Decoding Chain-of-Thought (VD-CoT) is proposed to decouple the decoding procedure into two steps. The model first predicts the type and number of visual entities that are going to be decoded, and then perform the actual decoding. A large collection of visual instruction tuning data, VT-Instruct, is constructed from multiple sources covering various visual tasks.

**Strengths:**

1. The multimodal encoding and decoding plugins enable solving various vision-language tasks in a unified model.

2. The idea of VD-CoT is interesting, which requires the model to "think internally" and decide which visual units are going to be decoded in the actual response. This could be helpful for improving the decoding accuracy.

**Weaknesses:**

1. Although the model is able to perform various tasks, the performance in each domain is questionable. In the quantitative results, this paper seems to be deliberately comparing with weaker models. Examples include but are not limited to:
    - Table 2: Only earlier grounded MLLMs are compared. An early baseline, LLaVA-1.5-7B (the "fundamental MLLM" module in the proposed method should be comparable to it) can achieve 78.5 accuracy on VQAv2.
    - Table 3: On POPE, "F1 score" should be the main metric to compare with, as one may achieve a very high precision by making the model predicting conservatively. Previous models like LLaVA-1.5 also mainly reports the F1 score.
    - Table 4: The performance on REC and RES are clearly behind more recent models. For example, GLaMM [ref1] achieves 83.2 RES cIoU on RefCOCO TestA, and UNINEXT [ref2] achieves 89.4 REC accuracy (IoU>0.5) on RefCOCOg Test.

2. Although this work claims keypoints as one major decoding modality, there is no quantitative evaluation of keypoint detection.

3. The ablation study does not discuss the most important module designs, e.g., decoding with CoT vs. without CoT, how data mixture in VT-Instruct affects the final results.

4. The presentation is not suitable for an academic research paper. In sections 3 and 4, only a high-level overview of the method is introduced, with a few short paragraphs and 5 large-sized figures. After reading these sections, readers may get a glimpse of the model and data, but too many details are missing or hidden in the appendix.

5. The qualitative results mainly show simple cases with few objects clearly visible. Even so, there seems to be many artifacts in the visualization (e.g., duplicate "person" detections in Figure 8 top right, low-quality masks in Figure 9 top right). It is unclear how the model perform in more challenging natural scene images.

[ref1] Hanoona Rasheed, Muhammad Maaz, Sahal Shaji, Abdelrahman Shaker, Salman Khan, Hisham Cholakkal, Rao M. Anwer, Eric Xing, Ming-Hsuan Yang, Fahad S. Khan. GLaMM: Pixel Grounding Large Multimodal Model. In CVPR, 2024.

[ref2] Bin Yan, Yi Jiang, Jiannan Wu, Dong Wang, Ping Luo, Zehuan Yuan, Huchuan Lu. Universal Instance Perception as Object Discovery and Retrieval. In CVPR, 2023.

**Questions:**

Please check the weakness section.

---

### Official Review · Reviewer_Lrjh · 2024-11-04

**Soundness:** 2
**Presentation:** 3
**Contribution:** 2
**Rating:** 5
**Confidence:** 5

**Summary:**

This paper aims to enhance the MLLM to solve dense visual prediction tasks. The authors first propose the VD-CoT to provide the necesssary phrase, unit and learnable queries for the decoders. Then, they introduce the VT-PLUG network by incorporating MLLM with several task-specific meta decoders and supports for visual/text prompts inputs. To the proposed model, they collect the large-scale VT-Instruct dataset. The authors comprehensively validate the method in various visual tasks such as visual understanding, REC, GCG, etal.

**Strengths:**

Pros:
1. The paper proposes the VD-CoT technique to select the proper decoder for completing differnt tasks.
2. The paper collects the large-scale VT-Instruct and make them into the same format to support various visual tasks.
3. The authors conduct the comprehensive experiments on various visual tasks, such as visual understanding, REC, GCG.
4. The authors provides the code, which would benifit the research community.

**Weaknesses:**

Cons:

1. The related works lack a comprehensive review of literature on enhancing MLLMs with visual prediction capabilities. Key examples include the text-based method VITRON [1], and end-to-end methods like Next-GPT [2], VisionLLM series [3,4]. I think it is better to move the Sec.4.1 to the related works.
2. The process of VD-CoT is unclear to me. When is VD-CoT performed? How is the model supervised with VD-CoT during training? In Figure 2, when the user prompt requests a visual task, does the model directly output the answer?
3. The statement of group hungarian matcher is very brief in Sec.4.2, while there is the ablation study of it. Could you provide more details about group hungarian matcher and its difference with standard hungarian matcher?
4. From Table 4 and examples in Appendix, It seems that the performance of segmentation is not well. What is the reason? Did the authors train the model on the segmentation tasks only?
5. Why the VD-CoT would prevent task conflict? As to me, it is a technique to select the proper decoder and give the semantic learnable queries for decoders. The performance conflict between different decoders is not well studies.
6. The paradigm of integrating MLLMs with multiple meta decoders has been explored in studies [2,4]. I would like to hear from the authors about the differences between this submission and previous methods.



[1] Fei, H., Wu, S., Zhang, H., Chua, T. S., & Yan, S. (2024). VITRON: A Unified Pixel-level Vision LLM for Understanding, Generating, Segmenting, Editing.
[2] Wu, S., Fei, H., Qu, L., Ji, W., & Chua, T. S. (2023). Next-gpt: Any-to-any multimodal llm. arXiv preprint arXiv:2309.05519.
[3] Wang, W., Chen, Z., Chen, X., Wu, J., Zhu, X., Zeng, G., ... & Dai, J. (2024). Visionllm: Large language model is also an open-ended decoder for vision-centric tasks. Advances in Neural Information Processing Systems, 36.
[4] Wu, J., Zhong, M., Xing, S., Lai, Z., Liu, Z., Wang, W., ... & Dai, J. (2024). VisionLLM v2: An End-to-End Generalist Multimodal Large Language Model for Hundreds of Vision-Language Tasks. arXiv preprint arXiv:2406.08394.

**Questions:**

Some advice and questions that would not influence the decision.
1. In Figure 2, attaching the original image would enhance the understanding of the VD-CoT example.
2. It is better to list a table for the training details, including batch size, epoch, lr, etal.
3. It seems that in each response, only one of the decoders will be invoked. Can the model complete multiple tasks in a single response, such as segmenting and providing keypoints simultaneously?
4. Regarding the training details, since the model includes several meta decoders, some decoders might not be utilized in each training iteration, which would make the training hang up. How do you solve the problem?

---

### Official Review · Reviewer_RWvb · 2024-11-04

**Soundness:** 2
**Presentation:** 2
**Contribution:** 2
**Rating:** 3
**Confidence:** 4

**Summary:**

This work proposed a framework that centered with MLLM and use visual plugins to complete visual tasks such as grounding and segmentation.
To achieve satisfactory results, it propose a VT-PLUG framework and build a VT-Instruct dataset. Proposed framework achieves improved performance on most of downstream tasks.

**Strengths:**

This work unifies multiple vision tasks into a unified framework and use a framework of plugins to complete tasks.

**Weaknesses:**

1. Proposed methods use existing expect tools to decoder visual predictions, which is unfair to compare with previous MLLMs. Such as Shikra has no decoder stage after LLM decoder.
2. As MLLM in the proposed framework works as a router and generator for inputs to downstream decoders, I think this work is closer to ones using MLLM as an agent but not an end-to-end model. Thus, maybe it should be compared with works like huggingGPT.
3. Organization of this paper is somehow poor and hard to follow. I think putting the overview figure for the whole pipeline starting from input images and ending with target predictions would be better. Definition of plugins can be explained in further details.

Shen, Yongliang, et al. "Hugginggpt: Solving ai tasks with chatgpt and its friends in hugging face." Advances in Neural Information Processing Systems 36 (2024).

**Questions:**

Please see in Weakness.

**Details Of Ethics Concerns:**

This work use existing academic datasets to generate instruction-following dataset.

---

### Note · Authors · 2024-11-15

I have read and agree with the venue's withdrawal policy on behalf of myself and my co-authors.